# Functional Adhesion of Pectin Biopolymers to the Lung Visceral Pleura

**DOI:** 10.3390/polym13172976

**Published:** 2021-09-02

**Authors:** Yifan Zheng, Aidan F. Pierce, Willi L. Wagner, Hassan A. Khalil, Zi Chen, Andrew B. Servais, Maximilian Ackermann, Steven J. Mentzer

**Affiliations:** 1Laboratory of Adaptive and Regenerative Biology, Brigham & Women’s Hospital, Harvard Medical School, Boston, MA 02115, USA; YZHENG@BWH.HARVARD.EDU (Y.Z.); afpierce@bwh.harvard.edu (A.F.P.); willi.wagner@uni-heidelberg.de (W.L.W.); hakhalil@bwh.harvard.edu (H.A.K.); zchen33@bwh.harvard.edu (Z.C.); aservais@bwh.harvard.edu (A.B.S.); 2Department of Diagnostic and Interventional Radiology, Translational Lung Research Center, University of Heidelberg, 69120 Heidelberg, Germany; 3Institute of Functional and Clinical Anatomy, University Medical Center of the Johannes Gutenberg-University, 55131 Mainz, Germany; maximilian.ackermann@uni-mainz.de

**Keywords:** lung, air leaks, pleura, pectin, adhesion, polysaccharide, scanning electron microscopy

## Abstract

Pleural injuries and the associated “air leak” are the most common complications after pulmonary surgery. Air leaks are the primary reason for prolonged chest tube use and increased hospital length of stay. Pectin, a plant-derived heteropolysaccharide, has been shown to be an air-tight sealant of pulmonary air leaks. Here, we investigate the morphologic and mechanical properties of pectin adhesion to the visceral pleural surface of the lung. After the application of high-methoxyl citrus pectin films to the murine lung, we used scanning electron microscopy to demonstrate intimate binding to the lung surface. To quantitatively assess pectin adhesion to the pleural surface, we used a custom adhesion test with force, distance, and time recordings. These assays demonstrated that pectin–glycocalyceal tensile adhesive strength was greater than nanocellulose fiber films or pressure-sensitive adhesives (*p* < 0.001). Simultaneous videomicroscopy recordings demonstrated that pectin–glycocalyceal adhesion was also stronger than the submesothelial connective tissue as avulsed surface remnants were visualized on the separated pectin films. Finally, pleural abrasion and hyaluronidase enzyme digestion confirmed that pectin binding was dependent on the pleural glycocalyx (*p* < 0.001). The results indicate that high methoxyl citrus pectin is a promising sealant for the treatment of pleural lung injuries.

## 1. Introduction

The most common disease treated by thoracic surgery is lung cancer. Worldwide, more than 2 million people were diagnosed with lung cancer in 2018 [1,2,3]. Thoracic surgery is also required for the removal of benign lung tumors, the diagnosis of inflammatory lung diseases, and the removal of pleural and chest wall tumors [4]. Injury to the surface of the lung—the delicate visceral pleura—is the most common complication after thoracic surgery [5,6,7,8].

The consequence of surgically induced, spontaneous, or even traumati c disruption of the pleura is an “air leak.” Air leaking out of the lung is the result of transpulmonary pressure; that is, the pressure difference between the pleura outside the lung and airways inside the lung [5]. In most patients with pleural injury, transpulmonary pressure results in the intrapleural accumulation of air called a pneumothorax. Depending upon on the physiologic conditions and nature of the injury, the pneumothorax may be stable or progressive. Ongoing expansion of the pneumothorax can result in life-threatening collapse of the lung. Accumulation of the pneumothorax under pressure—a tension pneumothorax—may lead to cardiovascular collapse as well [5].

In addition to life-threatening consequences, air leaks are the primary reason for increased hospital length of stay (LOS) after pulmonary surgery [9,10]. Pleural air leaks increase LOS by 5 to 13 days, more than doubling the cost of hospitalization [11]. Moreover, the impact of air leaks on patient recovery and hospital resources is significant. Economic analyses indicate that air leaks after pulmonary surgery lead to additional complications in both the lung and pleural space, such as atelectasis, pneumonia, empyema, and the need for chest drains [12]. Post-surgical prolonged air leaks increase the rate of readmission within 30 days by 20.4% [13].

A plant-derived biopolymer called pectin, edible in jams and jellies [14], has been proposed as a potential patch-sealant of pleural air leaks [15,16,17]. Pectin is a structural heteropolysaccharide that is comprised of as much as 30% of the primary cell walls of plants [18]. Pectin mainly consists of esterified d-galacturonic acid residues in (1→4) chains [19,20]. Importantly, pectin has a free volume that facilitates the embedding of drugs or other substances within the polymer [21]. The bioadhesivity of pectin, combined with the ability to embed drugs or growth factors within the gel structure, has led to considerable interest in using pectin to target drug delivery [22] and facilitate wound healing [23].

In our previous work, we have used pressure-decay testing—a common method in non-biologic product testing [24]—as a method to evaluate the sealant function of pectin [25]. The pressure-decay method pressurizes the test system until it reaches a set pressure and then monitors decay. We have used this global test of sealant function to demonstrate that citrus pectin films are effective at producing air-tight seals of the mouse lung [17], human pleura [16], and swine lung [26]. Furthermore, airway contiguity facilitates forced-oscillation impedance testing to evaluate the restrictive (entrapment) effects of the sealant on the lung surface. The forced-oscillation technique has demonstrated that acrylates produce significant lung restriction, whereas pectin is statistically indistinguishable from untreated control lungs [17]. Despite these encouraging observations of global lung function, our understanding of pectin adhesion at the pleural interface remains limited. 

In this report, we investigated the functional bioadhesion between high-methoxyl pectin and the visceral pleura surface of the lung. The practical limitations of bioadhesion testing were illustrated by high resolution imaging of the adhesive interface. 

## 2. Methods

**Animals.** Male mice, 35 gm wild-type C57BL/6 (Jackson Laboratory, Bar Harbor, ME,), were anesthetized prior to euthanasia. The care of the mice was consistent with guidelines of the American Association for Accreditation of Laboratory Animal Care (Bethesda, MD, USA) and was approved by the Brigham and Women’s Hospital Institutional Animal Care and Use Committee. Pleural adhesion assays were performed with porcine small bowel, which was procured by a local vendor (Research 86, Boston, MA, USA), and studied with a protocol approved by the Brigham and Women’s Hospital Institutional Animal Care and Use Committee.

**Pectin.** The unstandardized citrus pectins used in this study were obtained from a commercial source (Cargill, Minneapolis, MN, USA). The characterization of the high methoxyl citrus pectin has been detailed elsewhere [27]. Briefly, the proportion of galacturonic acid residues in the methyl ester form determines the degree of methoxylation. The high-methoxyl pectins (HMP) demonstrated a greater than 50% degree of methoxylation. The pectin powder was stored in low humidity at 25 °C.

**Pectin dissolution in water.** The pectin powder was dissolved at 25 °C by a controlled increase in added water to avoid undissolved powder [28]. Sequential swelling, softening, and fluidization of the particles was followed by dissolution [29]. As described elsewhere [30], the complete dissolution of the pectin was produced using a high-shear 10,000 rpm rotor-stator mixer (L5M-A, Silverson, East Longmeadow, MA, USA). Reproducible viscosity was ensured using a digital tachometer and ammeter (DataLogger, Silverson). The dissolved pectin was poured into variable-sized molds for further studies.

**Pressure-sensitive adhesive (PSA).** The PSA was a proprietary multi-purpose acrylic adhesive made available through the cooperation of the 3M Corporate Research Materials Laboratory (St. Paul, MN, USA).

**Nanocellulose fibers (NCF).** The NCF powder, obtained from the University of Maine (Process Development Center, Orono, ME, USA), was dissolved at 25 °C by a controlled increase in water similar to previous reports. Briefly, NCF dissolution was obtained with progressive hydration, followed by a high-shear 10,000 rpm rotor-stator mixer (L5M-A, Silverson). The dissolved NCF was poured into standardized molds and cured for further studies.

**Adhesion testing.** Pectin–lung adhesion experiments were performed with a custom fixture designed for the TA-XT plus with 5 kg of the load cell (Stable Micro Systems). The fixture was composed of a 30 mm diameter flat-ended stainless-steel cylindrical probe and a flat stainless-steel fixture surface; both surfaces used adhesive mounts to fix the pectin film and tissue specimen. The probe descended at the selected probe velocity until encountering a trigger force of 1N. The probe compressed the pectin films and pleural sample at a selectable compression force (typically 1–5N) and development time. The probe was then withdrawn at 0.5 mm/s with constant force and distance recordings at 500 pps.

**Transillumination stereo microscopy.** The film interface was transilluminated with a 4000 lumen 6000 K LED light with custom diffusion filter to assure uniform illumination. Probe compression and withdrawal was recorded with a 16 Mega pixel camera (Hayear, Shenzhen, PRC) at 60 frames per second mounted to a Nikon SMZ 1000 stereo microscope (Nikon, Tokyo, Japan) as previously described [31]. The recorded MOV files were converted to MetaMorph (Molecular Devices, Downington, PA, USA) compatible STK files for morphometric analysis. Time-base correction was integrated into the recordings for calibration of the image stacks.

**Scanning electron microscopy.** After coating with 20–25 A gold in an argon atmosphere, the pectin films were imaged using a Philips XL30 ESEM scanning electron microscope (Philips, Eindhoven, the Netherlands) at 15 Kev and 21 μA. A eucentric sample holder was used for standardized automation.

**Enzyme treatment**. The tissues were treated with three commercially obtained enzymes previously used (Sigma-Aldrich, St. Louis, MO, USA). Hyaluronidase cleaved the 1/4 linkages between N-acetyl-D-glucosamine and D-glucuronate. The hyaluronidase solution and tissues were maintained at 37 °C during a 90-min incubation. After enzyme treatment, the tissues were washed with PBS three times.

**Statistical analysis.** The statistical analysis was based on measurements in at least three different samples. The unpaired Student’s *t*-test for samples of unequal variances was used to calculate statistical significance. The data was expressed as mean ± one standard deviation. The significance level for the sample distribution was defined as *p* < 0.01.

## 3. Results

**Pectin adhesion to the pleura.** The normal mesothelium is composed of a polygonal epithelial layer (Figure 1A) covered by microvilli of varying lengths and densities (Figure 1B). When a glass phase pectin film (80 um thick) was applied to the mesothelium with gentle pressure (3–5 N) for 20 s, the pectin appeared to fuse with the mesothelium. Subsequent scanning electron microscopy (SEM) of the lung surface showed that the adherent pectin created a smooth surface; that is, pectin effectively obscured the microvilli and cobblestone appearance of the original mesothelium (Figure 1C,D). Areas of non-adherent pectin demonstrated a cribiform appearance (Figure 1C), suggestive of dissolution. Evidence of marginal stranding was also seen (Figure 1D–F).

**Pectin–glycocalyceal adhesion.** To quantitatively assess pectin adhesion to the pleural surface, we developed a tensile adhesion strength assay (Figure 2). A cylindrical probe with force, distance, and time recordings was used to gently compress the pectin and pleural samples for 60 s, followed by probe withdrawal. The separation of the pectin and pleura was recorded by synchronized videomicroscopy. Despite very thin (1 mm) pleural samples, the pleural tissue demonstrated remarkable elasticity. Probe withdrawal resulted in the pleural surface stretching several millimeters. Videomicroscopy demonstrated a corresponding increase in optical transmittance (Figure 3). The tensile adhesion strength of the branched-chain pectin was significantly greater than linear-chain nanocellulose fibers (NCF) and industrial pressure-sensitive adhesives (PSA) (Figure 4A–C). Consistent with tissue elasticity, pectin–pleural adhesion demonstrated a blunted peak force and prolonged withdrawal force. These physical characteristics resulted in a significantly increased work of adhesion (*p* < 0.001) (Figure 4D).

**Submesothelial separation.** The dynamics of the adhesion assay were notable for submesothelial tissue separation, as seen on videomicroscopy. The separation was visualized as subpleural lucent regions (Figure 5A, ellipse). In addition, the assay was frequently associated with the residual tissue remaining adherent to the pectin (see also Figure 3A). To investigate the morphologic consequences of subpleural tissue failure, we performed scanning electron microscopy of the pectin surface. SEM demonstrated tissue separation at the level of subpleural alveoli. The remnant lung tissue frequently demonstrated core structural features, such as the central line element [32], as being bound to the pectin interface (Figure 5C). In addition, it is noteworthy that there were apparent examples of pectin–glycocalyceal adhesions at the margins of the avulsed tissue (Figure 5D).

**Glycocalyx-dependent adhesion.** To test the dependence of pectin–pleural adhesion on the glycocalyx, we used both enzymatic digestion and mechanical abrasion to remove the glycocalyx (Figure 6). Hyaluronidase, an enzyme known to remove the glycocalyx [33], was used to pre-treat the pleura. A representative sample shows the pleura from the same lung pretreated with hyaluronidase (blue line). Pleural adhesion was significantly lower than the untreated lung (red line). Triplicate samples demonstrated a significant decrease in the work of adhesion (*p* < 0.001). Similarly, mechanical abrasion of the glycocalyx resulted in a significant decrease in the work of adhesion (*p* < 0.001).

## 4. Discussion

In this report, we have demonstrated the morphologic and mechanical properties of pectin adhesion to the visceral pleural surface of the lung. First, we show morphologic evidence that high-methoxyl citrus pectin films intimately bind to the lung surface. Second, we demonstrate that pectin–glycocalyceal adhesion is stronger than submesothelial tissue, NCF, or PSA. Third, mechanical abrasion and enzymatic digestion demonstrated that pectin adhesion strength is dependent upon the pleural glycocalyx. These results indicate that high methoxyl citrus pectin is a promising sealant for the treatment of pleural lung injuries.

The normal mesothelium is a polygonal cell layer with a dense forest of microvilli expressed on the cell surface [34,35]. The obscuration of the mesothelial microvilli with pectin binding provides a convenient measure of the topography of pectin adhesion. SEM demonstrated that pectin broadly covered the pleura and conformed to the contours of the lung surface. The appearance of the pectin suggested that it had fused into the mesothelial surface, a finding consistent with pectin being an air-tight sealant [16]. 

The strength of pectin–pleural adhesion proved to be surprisingly difficult to measure because of the elasticity of the tissue subjacent to the pleura. The tensile adhesion strength assay more accurately reflected the mechanical properties of the submesothelial tissue than the strength of the pectin–pleural adhesion. Tissue remnants, indicating preserved pectin–pleural adhesions, were commonly visualized on the pectin film after probe separation. Despite these quantitative limitations, our adhesion assay provided insight into the relative strength of pectin–pleural adhesion. Pectin demonstrated significantly greater adhesion to the lung surface than either linear-chain polysaccharides (e.g., NCF) or more traditional pressure-sensitive adhesives.

Pectin adhesion to the lung surface reflects the function of pectin in plants. Pectin is the glue in the middle lamella of plants [36]; that is, pectin is responsible for the tensile strength of tall trees, as well as the shear resistance of plants in tropical force winds [37]. Pectin adhesion appears to be the result of the entanglement or interpenetration of pectin chains and cellulose microfibrils [38]. The evidence for pectin entanglement is the diminished adhesion after the covalent modification or the restrictive crosslinking of pectin chains [39]. Analogous to commercial hook-and-loop Velcro, pectin entanglement provides a physical mechanism of adhesion that is relatively independent of the stringent reaction conditions of covalent chemistry. Moreover, the physical process of entanglement provides both the adhesive strength as well as the controlled slippage necessary to accommodate plant growth. 

Finally, our findings suggest that both the endogenous pleural glycocalyx and surface-bound pectin films may play an important role in healing lung injuries. Injury to the visceral pleural glycocalyx―common in routine lung surgery―is associated with slow healing and prolonged hospitalization [13]. To facilitate surface repair, pulmonary surgeons have empirically recognized the value of buttressing visceral pleural injuries with neighboring parietal pleura. A variety of surgical techniques, including pleural flaps [40] and pleural tents [6], have been designed to facilitate the healing of pleural injures. Due to the structural biosimilarity of pectin and the pleural glycocalyx [41], we speculate that pectin may have a similar function in facilitating visceral pleural repair.

## Figures and Tables

**Figure 1 polymers-13-02976-f001:**
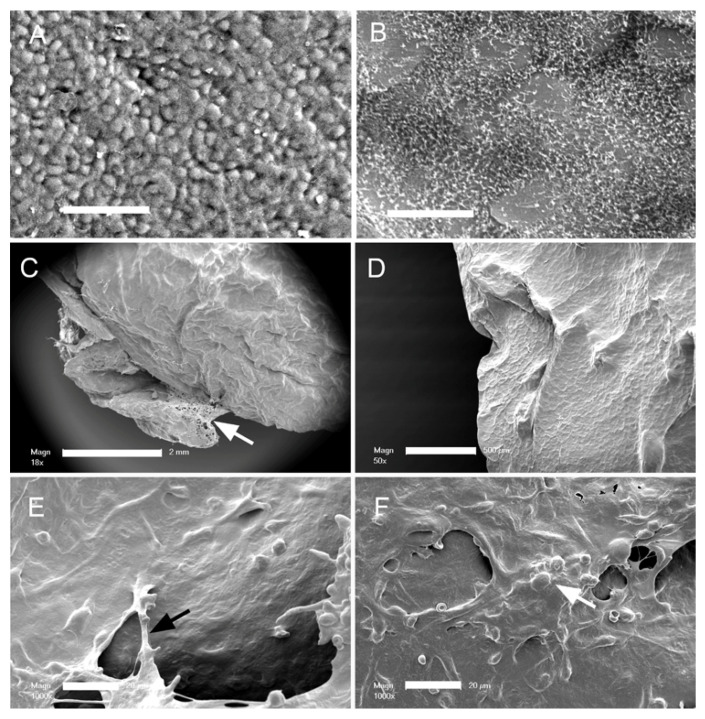
Scanning electron microscopy of the murine visceral pleural surface before and after the application of the pectin bioadhesive. (**A**) Normal visceral pleural surface with cobblestone mesothelial cells (bar = 50 um). (**B**) The dense microvilli on the mesothelial cell surface is visible with higher magnification (bar = 20 um). (**C**,**D**) After application of the pectin film, there is a notable change in the surface morphology, with the microvilli apparently covered by the pectin film. Note the pectin adapting to the contour of the pleural surface ((**C**), bar = 2 mm; (**D**), bar = 500 um). The unbound pectin developed a cribiform pattern (arrow), suggesting dissolution of the unbound pectin (**E**,**F**). Higher magnification of the pectin film showed evidence of fibrillation or stranding between the bound pectin film and unbound films (white arrow), as well as molding around mesothelial cell nuclei (black arrow) ((**D**,**E**) bar = 20 um).

**Figure 2 polymers-13-02976-f002:**
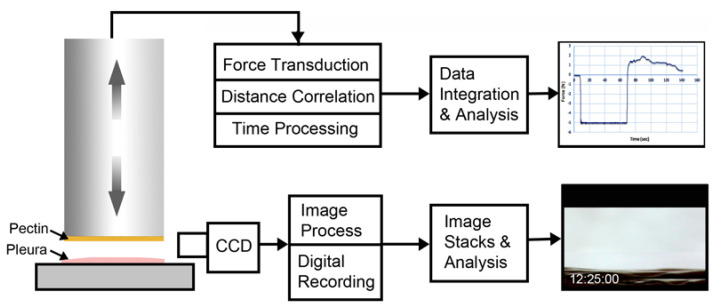
Schematic of pectin–glycocalyceal adhesion testing. The material analyzer (Stable Micro Systems) used custom fixtures to mount pectin films on the cylindrical probe and visceral pleural to the fixture surface. The computer-controlled adhesion test involved probe descent at 2 mm/s until it reached a trigger force of 5N. The 5N force was maintained for a 60-s development time. The probe was then withdrawn at a velocity of 0.2 mm/s. Time, force, and distance were recorded at 500 points per second (pps) and analyzed using the Exponent 6.0 software (Stable Micro Systems) with custom scripts. Due to the complex adhesion process and extensibility of the pleura, simultaneous time-base corrected digital videomicroscopy was performed of the adhesion interface and analyzed using MetaMorph 7.1 software (Molecular Devices).

**Figure 3 polymers-13-02976-f003:**
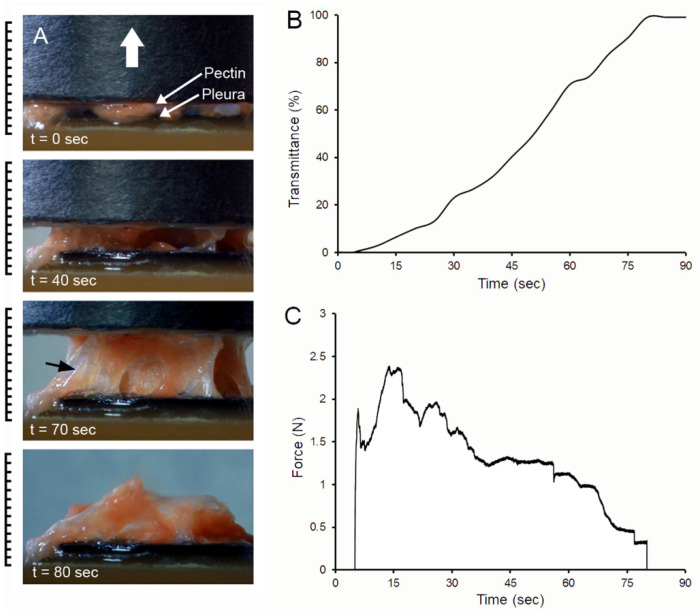
Probe withdrawal in the pectin–pleura adhesion test with simultaneous force–distance and videomicroscopy recordings. (**A**) The pectin film and a corresponding 1 mm thick patch of porcine visceral pleura were compressed at 5N for 60 s prior to probe withdrawal; the direction of the cylindrical probe movement (0.5 mm/s) is indicated by the white arrow. Separation of the surfaces is associated with submesothelial tissue rupture (black arrow). At the conclusion of the withdrawal, residual tissue can be observed adhering to the pectin film. (**B**) Progressive transillumination of the interface is shown by a percentage of optical transmittance. (**C**) The force required to maintain the probe velocity (0.5 mm/s) is shown.

**Figure 4 polymers-13-02976-f004:**
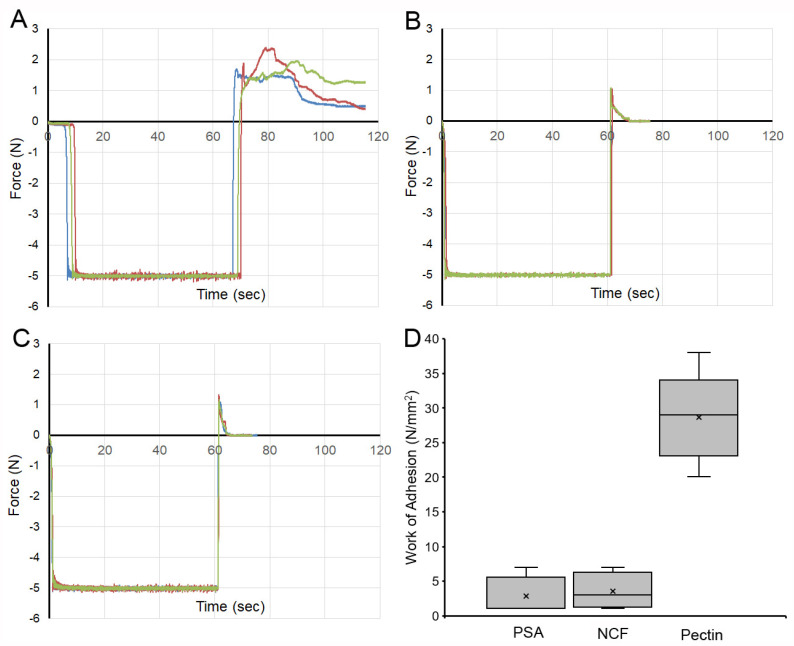
Comparison of the adhesion strength of pectin, nanocellulose fibers (NCF), and pressure-sensitive adhesive (PSA) to the porcine visceral pleura. In each condition, the pleural interface was compressed by the probe at 5N pressure for 60 s. The force required for probe withdrawal at 0.5 mm/s was recorded at 500 pps. (**A**) Pectin films and representative force tracings of probe withdrawal. NCF films (**B**) and PSA (**C**) demonstrate lower peak adhesion and rapid adhesion failure. (**D**) The work of adhesion, defined as the area under the force–distance curve, demonstrated a significant increase for pectin compared to NCF and PSA (*p* < 0.001). The box spans the interquartile range with the median marked with an X and the whiskers defining the data range.

**Figure 5 polymers-13-02976-f005:**
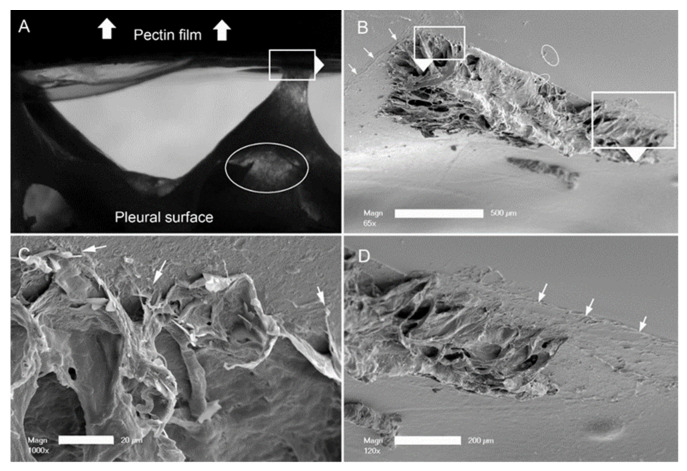
Pectin–glycocalyceal adhesion resulting in submesothelial tissue separation with the result of porcine lung tissue adherent to the pectin film. (**A**) Adhesion assay with a representative example of submesothelial separation, reflected in multiple areas of the lucent tissue (ellipse). The direction of the probe withdrawal is denoted by the white arrow. (**B**) The tissue separation in the subjacent alveoli. Higher resolution images show connections of the central line elements of the lung adherent to the pectin (**C**) and residual areas of pectin–glycocalyceal adhesion can be visualized ((**D**), arrows) ((**B**), bar = 500 um; (**C**), bar = 20 um; and (**D**), bar = 200 um).

**Figure 6 polymers-13-02976-f006:**
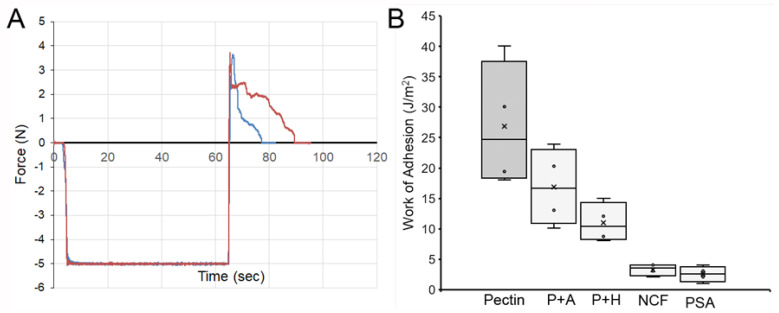
Mechanism of pectin adhesion to the porcine pleural glycocalyx. (**A**) Pectin adhesion to the native pleural surface (red line) and after enzyme treatment with hyaluronidase (blue line). The area under the curves reflect the work of adhesion. (**B**) Comparison of pectin adhesion to the native glycocalyx (dark gray) compared to the glycocalyx diminished by pleural abrasion (P + A) and hyaluronidase (P + H) pre-treatment (light gray). Nanocellulose fibers (NCF) and pressure-sensitive adhesive (PSA) are also shown for comparison. The box spans the interquartile range of the replicate samples with the median marked with an X and the whiskers defining the data range.

## Data Availability

Not applicable.

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
