# Peer review of "Functional Adhesion of Pectin Biopolymers to the Lung Visceral Pleura"

_polymers, 2021, doi:10.3390/polym13172976_

Round 1

Reviewer 1 Report

I have the following comments:

  1. The introduction part is very short. Some information about the use of pectin should be added, same as information about the use of pectin for the medical purposes. The following reference can be used: Dordevic, D., Jancikova, S., Capikova, J., Tremlová, B., & Kushkevych, I. (2020). Chemical and sensory properties of fruit jams affected by bamboo fiber fortification.
  2. It is not written what kind of lung was used. The material and methods part is also written very scarcely.
  3. The conclusion is also not written.

The whole manuscript is giving very little information and it should be thoroughly rewritten.

Author Response

A1: We thank the Reviewer for the suggestion. We have significantly expanded the Introduction and included the recommended reference.

A2: We have expanded the Methods section and included appropriate animal information.

A3: We have also expanded the Conclusion section including a discussion of the structural complexity of pectin and its mode of adhesion; namely, entanglement.

A4: We recognize that the focus of this manuscript is different from most biochemical analyses; that is, we are analyzing the functional interaction of the biopolymer with the surface of the lung. We believe this novel perspective fills a gap in our current knowledge and will be valuable to other investigators.

Reviewer 2 Report

The authors solve the morphologic and mechanical properties of pectin
adhesion to the visceral pleural surface of the lung. To quantitatively assess
pectin adhesion to the pleural surface, authors used a custom adhesion test
with force, distance and time recordings.
The authors should devote more effort
to the description and analysis of the researched phenomenon-pectin adhesion
principle at the elementary/molecular level.  Also, comparison with other
materials of the same applications
(nanocellulose fiber films or pressure-sensitive adhesives) should
be processed on more fundamental/functional level.

Author Response

A1:  We appreciate the Reviewer's perspective and recognize the unique translational nature of this manuscript.  Here, we describe the functional realities and molecule interactions at the pectin-pleural interface.  We believe this physical context is crucial for the interpretation of ongoing molecular studies.  No prior work in this area is available.  We have modified the manuscript to address more elementary observations where possible.

A2:  We agree with the Reviewer in principle.  We need more fundamental investigations of the pectin interaction at the pleural surface.  This manuscript provides the important groundwork for these future studies.  We note that the scanning electron micrographs represent the state-of-the-art of SEM imaging of biopolymers.  We believe this data, showing physical events at the pleural surface, will be indispensable for the interpretation and application of future molecular studies.